# Production and Physical–Mechanical Characterization of Peat Moss (Sphagnum) Insulation Panels

**DOI:** 10.3390/ma14216601

**Published:** 2021-11-02

**Authors:** Günther Kain, Marco Morandini, Angela Stamminger, Thomas Granig, Eugenia Mariana Tudor, Thomas Schnabel, Alexander Petutschnigg

**Affiliations:** 1Department for Forest Products Technology and Timber Construction, Salzburg University of Applied Sciences, Markt 136a, 5431 Kuchl, Austria; gkain.lba@fh-salzburg.ac.at (G.K.); astamminger.htw-m2019@fh-salzburg.ac.at (A.S.); tgranig.htw-m2019@fh-salzburg.ac.at (T.G.); eugenia.tudor@fh-salzburg.ac.at (E.M.T.); thomas.schnabel@fh-salzburg.ac.at (T.S.); alexander.petutschnigg@fh-salzburg.ac.at (A.P.); 2Department for Wood Restoration Technology, Higher Technical College Hallstatt, Lahnstraße 69, 4830 Hallstatt, Austria; 3Salzburg Center for Smart Materials, c/o Department of Chemistry and Physics of Materials, Paris Lodron University of Salzburg, Jakob-Haringer-Strasse 2a, 5020 Salzburg, Austria; 4Faculty of Furniture Design and Wood Engineering, Transilvania University of Brasov, B-dul. Eroilor nr. 29, 500036 Brasov, Romania

**Keywords:** peat moss, insulation materials, natural materials, renewable materials

## Abstract

Peat moss (sphagnum) is a commonly used sealant, fill, and insulation material in the past. During the efforts to rewet drained moors due to ecological considerations, the technical use of peat moss (sphagnum farming) again became the focus of attention. In the framework of this investigation, insulation panels consisting of peat moss, bound with urea formaldehyde, were produced. Panels manufactured in a wet process and mats bound with textiles were also fabricated. The specimens’ thermal conductivity, water vapor diffusion resistance, modulus of rupture, modulus of elasticity, internal bond, compression resistance, water absorption, and thickness swelling were measured. Physical–mechanical properties were adequate with the resin-bound panels, but not with wet process panels. Moss mats had good characteristics for cavity insulation purposes. The thermal conductivity of the moss panels and mats was found to be lowest with a density of 50 kg/m^3^, accounting for 0.04 W/m·K. The results show that peat moss is a promising resource for production insulation panels, because their thermal conductivity and mechanical stability are comparable to other insulation materials.

## 1. Introduction

The investigation of historic building constructions not only yields information about traditional work technologies, but also brings to light materials which have been preserved over a long period. Under real-life conditions, the performance and functionality of building materials and building constructions can be evaluated. From this point of view, the building cultural heritage represents a long-term experiment with integrated experience knowledge transferred from generation to generation. Starting from the mid of the 20th century, this tradition ended, and industrial building materials are now applied areawide.

The sealing of seams in historic wood constructions seems to be of interest from this point of view, because the demand for ecologic and durable gap sealing systems is increasing in the market segment of wooden building constructions. Peat moss (sphagnum) can be found in the gaps of historic wood constructions in the UNESCO world heritage region Hallstatt-Dachstein/Salzkammergut.

Peat moss grows in moist forests or on marsh areas, but can most frequently be found in high moors [1]. It grows under natural conditions, predominantly in acidic or sub-neutral, nutrient-poor environments with a high water level [2]. Peat moss is the most important factor for the growth of high moors [3], where moss has built peat layers several meters thick since the last glacier epoch [2,4]. Whereas peat moss was harvested from local, natural moors in former times, its sustainable cultivation is currently the focus of research activities.

The cultivation of peat moss in paludi cultures (“palus” latin for marsh or morass) has been investigated for several years. Globally, moors contain twice as much carbon as the biomass of all forests, although they only cover 3% of the world’s area [5]. Drained and degraded moors can be rewetted and are used in agriculture and silviculture. After rewetting, peat moos can be grown on the surface and harvested (sphagnum farming) after 4 years on average. The peat moss (Figure 1) grows again and can be harvested again after a few years. Current investigations on sphagnum farming suggest that 3–6 tons of dry matter can be harvested per hectare and year [6]. By rewetting and keeping the water level high, sphagnum farming areas are suitable to store carbon. The thermal conductivity of peat moss soil is primarily influenced by the moisture content of the moss layer and, to a lower extent, by layer thickness and density [7].

The use of moss in the UNESCO world heritage region Hallstatt-Dachstein/Salzkammergut can be verified by written references since the mid-18th century [8]. Local moss habitats, the use of moss for sealing wooden boats, and specific working techniques are described. The technique to seal boats, called “Schoppen”, can also be found in Bavaria [9]. Historic constructions predominantly had wedge-shaped joints, in which the moss was plugged positively. In the narrower inner side of the joint, a narrow, trapezoidal lath was first adjusted to avoid the push-through of moss. The moss was spun, yielding a seal-plate which was placed in the joint and densified with a broader trapezoidal push lath. The latter was fixed using a U-shaped bracket made from sheet steel. When this construction is subjected to water, the construction elements absorb moisture and swell. The construction’s geometry and the swelling behavior of the moss automatically regulate the degree of the joint’s tightness [10].

Peat moss was used not only for boat constructions, but also for hydraulic structures. The wall of wooden dams on the water side had to be waterproof. The joints of the wall were filled with moss, and a wooden lath was nailed on the top [11].

Before placing the dry moss in the joint, it was spun in a kind of seal-cord. This can be verified not only by oral evidence [12], but also by the finding of such moss seal-cords in the joints of an 18th century solid wood construction in Gosau, Upper Austria. Notably, the lifetime of the joint sealing is over 250 years. In the joints of solid wood constructions, moss was not only used as a sealant against water but also as barrier to draft and heat loss. The stacking ground of scantlings was smoothly shaped using an axe, whereas, on their outer side a wedge-like joint was considered, which could be tightened with peat moss [13].

The use of peat moss mats for insulation purposes was investigated showing that the moss has 18% equilibrium moisture content (EMC) in a climate of 20 °C and 65% humidity, while wood shows an EMC of only 12% under these conditions. The thermal conductivity of moss mats with a density from 40 to 140 kg/m^3^ ranges from 0.036 to 0.041 W/m·K [10]. Another study focused on insulation panels made from peat moss mixed with straw in order to increase panel stiffness. Sodium silicate was used as a binder. Panels with a density of 170 kg/m^3^ showed a thermal conductivity (TC) of 0.034 W/m·K [14]. The uncompacted peat moss with a density of 18 kg/m^3^ had a thermal conductivity 30% higher than that of mats with a density of 50 kg/m^3^ [10], because the mechanisms of convective heat transfer resulted in a higher overall thermal conductivity in very light fiber materials [15]. Focusing on the moss TC, it covered a range otherwise achieved with artificial organic or inorganic materials. Peat moss can be pressed into specific forms, as shown by a study focusing on closing the mounting gap of windows with wedges made from peat moss [10].

The temperature-dependent increase in TC for peat moss mats is in the range of mineral wool [16], amounting to 0.19 mW/m·K per kelvin increase in temperature. This is significantly lower than with wood wool or ecologic bark insulation panels [17].

Many similar natural materials (e.g., reed, straw, hay, hemp, corn stalks, cotton, and flax) were successfully tested at laboratory scale for thermal insulation applications [18], and it seems appropriate to test peat moss for the same purpose, especially due to its interesting ecological context. In the framework of this investigation, the production of flexible insulation mats and panels produced in a wet process, as well as resin-bound panels made from peat moss, and their building physical characterization are discussed.

## 2. Materials and Methods

The peat moss for the current investigation was supplied by Paludi Culture Ramsloh, Germany and conditioned in a constant climate of 22.5 °C and 64% humidity until mass constancy was reached. The oven-dried mass of the moss and the material moisture and water content were determined according to ÖNORM EN ISO 18134-2 [19].

The sorption behavior of the peat moss was measured following the procedure of ÖNORM EN ISO 12571 [20]. Three peat moss and three saw dust samples were dried and subjected to 35%, 63%, and 95% humidity at 20 °C. Each humidity step was held until mass constancy of the samples.

The loose bulk density of the moss was determined according to ÖNORM EN ISO 17828 [21].

### 2.1. Production of Resin-Bound Peat Moss Panels

The calculation of the fiber mass and the used resin are based on technical standards of the wood fiber production. To achieve the target density as precisely as possible, the panels were subjected to a specific conditioning cycle. The grown structure of the peat moss consists of a densely grown and tight network. To achieve an even distribution of the adhesive, the material has to be crushed beforehand. The time factor plays an important role in achieving the desired particle size.

A laboratory plough disc mixer of the brand ETM-WHB75m was used to mix the moss fibers with the binder–hardener mixture. As a binder, urea formaldehyde, type Prefere 10F102 (66% solid content, pH 8.3–9, viscosity 60–90 mPa·s) from Metadynea (Krems, Austria) was used. In the first step, the moss fibers were filled into the drum of the mixer.

Afterward, the binder was applied evenly into the mixing drum while the blender was running. Until further processing of the resin-bound moss, the fibers were stored at a temperature of 10 °C. The fibers were evenly distributed in the press mold. The insulation panels made of peat moss were pressed at a density of 100 kg/m^3^, 150 kg/m^3^, and 250 kg/m^3^ with a Höfer (Taiskirchen, Austria) HLOP 280 laboratory press with a temperature of 180 °C. The total pressing time was 5 min (press factor 15 s/mm). It should be noted that the press cycle consisted of three ventilation phases to continuously remove the moisture in the mats. The insulation boards were pressed to a size of 32 cm × 32 cm × 2 cm. The produced laboratory panels (Figure 2) were stored in a standard climate of 20 °C with 60% humidity.

### 2.2. Production of Wet Process Panels

The moss was wetted with 500% water with reference to dry moss mass using an aerosol. It was put in airtight plastic bags for 10 min to ensure that the water was fully absorbed by the moss. The moist moss was put in press molds (25 cm × 25 cm), and panels with a density of 50, 100, 150, and 200 kg/m^3^ were pressed without any resin addition. The panels remained in the press mold for 10 days and were then dried in a climate room at 22.5 °C and 65% humidity until mass constancy.

### 2.3. Production of Moss Mats

The moss was placed between two jute nets with a mesh size of 1 cm and a grammage of 1.96 g/cm^2^. The mat was sewed using a hemp string with a diameter of 1 mm. The fissures were applied crosswise with a mesh size of 5 cm. The mats were produced with a density of 15, 20, and 50 kg/m^3^ and a size of 25 cm × 25 cm × 2 cm. The design of experiment for the test panels is summarized in Table 1.

### 2.4. Physical–Mechanical Characterization

The modulus of rupture (MOR) and modulus of elasticity (MOE) were determined according to ÖNORM EN 12089 [22,23], and the internal bond (IB) was determined according to ÖNORM EN 319 [24]. Specimens with a size of 50 mm × 50 mm were tested for IB using aluminum blocks which were glued on the specimens’ surface and then pulled apart in a universal testing machine. The compression resistance of specimens was tested following the rules of ÖNORM EN 826 [25]. The resin-bound panels were tested at 10% compression, whereas wet process panels and moss mats were tested at 50% compression due to their soft structure. Specimens were tested for water absorption and thickness swelling following ÖNORM EN 29767 [26]. Measurements were conducted after 24 h of water immersion.

The thermal conductivity (TC) was measured according to ÖNORM EN 12667 [27] using a temperature difference of 15 °C at an average platen temperature of 10, 25, and 40 °C, and the vapor diffusion resistance (VDR) was determined according to ÖNORM EN ISO 12572 [28]. In this way, the sample was subjected to a water vapor pressure difference under constant temperature. Circular samples were mounted at the top of glass cups with a diameter of 100 mm. Silica gel (dry cup) and desalinated water (wet cup) were filled until 15 mm below the surface of the sample. These materials resulted in an air humidity of 0% and 100%, respectively. The cups with the samples were stored in an air-conditioned chamber (20 °C, 65% RH). Periodical weighing enabled determining the vapor stream per time unit which is proportional to the diffusion resistance.

### 2.5. Statistical Analysis

The statistical analysis of the data was performed using the software SPSS 18 (IBM, New York, NY, USA) applying univariate ANOVA to test the effect of the panel type and the covariate panel density on the panel characteristics. The explanatory power of the explanatory variables was assessed evaluating partial eta-squared values in ANOVA.

## 3. Results and Discussion

### 3.1. Bulk Density of Peat Moss

The peat moss showed an average bulk density of 13.1 (standard deviation (SD) = 0.8) kg/m^3^ at a moisture content of 22%. The test layout for the determination of the bulk density according to ÖNORM EN ISO 17828 [21] requires compaction of the material by concussion before measurement. In the case of peat moss, a very low amount of settling was observed. The bulk density of peat moss was considerably lower than that of dry straw (24 kg/m^3^) or switchgrass (49 kg/m^3^) [29]. The bulk density of wood fibers from various wood species after refining was investigated, yielding densities between 25 and 29 kg/m^3^ [30]. The fibers in the cited study had a length below 1.4 mm. The peat moss particles in the current study were coarser, incorporating particles with a length up to 50 mm. A higher proportion of coarse fibers resulted in interlacing of individual fibers, in the formation of air gaps, and in a lower bulk density [30]. The bulk density of the raw material affects the compression ratio (board density divided by bulk density of the raw material) in panel production. Considering a bulk density of 13 kg/m^3^ for peat moss, the compression ratio for panels with a density ranging from 50–250 kg/m^3^ lies between 3.8 and 19.2. In der production of wood fiber insulation boards and medium-density fiberboard, this compression ratio ranges between 5 and 7 [31].

### 3.2. Moisture and Sorption Behavior

The curve in Figure 3 reveals the sorption characteristics of peat moss compared to that of wood dust. Starting with oven-dried material, the moisture content increased to 11.6% (SD = 0.02%) and 6.5% (SD = 0.01%) in the first climate with moss and saw dust, respectively. In a climate of 60% humidity, the material moisture accounted for 19.6% (SD = 0.02%) and 12.0% (SD = 0.02%). In the moist climate of 95% humidity, the peat moss reached a moisture content of 51.1% (SD = 0.06%), whereas the saw dust moisture content was 23.27% (SD = 0.01%). The equilibrium moisture content of the peat moss was respectively 5.1%, 7.6%, and 27.8% higher than that of saw dust.

Focusing on the sorption rate, both materials reacted equally quickly upon humidity changes, but the sorption rate of peat moss was 2–3 times higher than that of wood.

The moisture uptake from moist air differed clearly from that of wood saw dust. Peat moss took up significantly more moisture per mass unit. While the sorption behavior of wood is well known and the equilibrium moisture content (EMC) was in the range of what the standard literature reports (12% at 20 °C/65% humidity) [32], there is less work focusing on peat moss in this respect. A study on sorption curves for wood fiber materials reported an EMC of 26% at 97% humidity [33], which is in the range of wood dust in the current study. An experimental study on the potentials of peat moss for insulation purposes confirmed a significantly higher EMC than wood [10].

### 3.3. MOR/MOE

The bending strength of the investigated moss panels ranged from 0.1 to 1.45 N/mm^2^ (Table 2). Moss mats were not investigated in this respect due to their flexible characteristic. The statistical model (*p* = 0.000) revealed that 94% of the variation in MOR can be explained by the density and the panel type; whereas the effect of density was significant (*p* = 0.000), that of panel type was not. The explanatory power of the panel density was highest, explaining 93% of the variation in MOR. The MOR increased by 0.8 N/mm^2^ with a density increase of 100 kg/m^3^.

Focusing on MOE, the statistical model was significant (*p* = 0.000) in explaining 95% of the variation in MOE. Furthermore, only the effect of density was significant (*p* = 0.000) with an explanatory power of 94%. The stiffness of the panels increased by 69 N/mm^2^ per 100 kg/m^3^ increase in density.

Other work focusing on light insulation materials reported similar bending strength to the peat moss panels of this study (e.g., low-density binderless particleboard from kenaf core, 100–300 kg/m^3^, MOR 0–2 N/mm^2^ [34], bagasse particleboard bonded with citric acid and sucrose, 300–500 kg/m^3^, MOR 1–7 N/mm^2^ [35]). Commercially available wood wool insulation panels (350–600 kg/m^3^) were characterized by an MOR of 0.4 to 1.0 N/mm^2^ [36], comparable to the P3 moss panels. The MOE of light bark insulation panels with a density of 250 kg/m^3^ accounted for approximately 50 N/mm^2^ [17], a value exceeded by the peat moss panels with the same density. A study dealing with the load-bearing capabilities of an insulation panel (250–1300 kg/m^3^) made from miscanthus bound with mineral binders found a bending strength of 1.5 N/mm^2^ for the lighter panels [37], comparable to the moss specimens.

Whilst the resin-bound moss panels were adequately stiff and strong with a density over 100 kg/m^3^, the wet process panels lacked both properties.

### 3.4. Internal Bond (IB)

The IB ranged from 0.001 to 0.027 N/mm^2^. The moss mats showed the highest (M2), whereas the wet process panels showed the lowest IB.

Its variation was 79% caused by different panel types and panel density. Both aspects were statistically significant (*p* = 0.000); whereas the effect of panel type explained 77% of IB variation, the density had a much lower explanatory power of 45%.

For the same density of 100 kg/m^3^, the IB of resin-bound panels was three times higher than that of the wet process panels. The mats showed the highest IB with 0.027 N/mm^2^ at a density of 20 kg/m^3^ (Table 3).

The IB of wood fiber insulation materials accounted for 0.003 to 0.05 N/mm^2^ [36], indicating that the proposed moss insulation panels made from peat moss showed adequate IB values. The moss mats with textile reinforcement incorporated the tensile strength via the jute net and the hemp cord, which influenced the IB to a great extent.

### 3.5. Compression Resistance (CR)

The compression resistance of the investigated samples ranged from 0.01 (M1) to 0.50 N/mm^2^ (P3) (Table 4). Specifically, 86% of this variation was explained by differing panel type (*p* = 0.001)) and density (*p* = 0.000). The explanatory power of the density was 76%, being significantly higher than that of the panel type with 50%.

The CR of the light wet process and mat specimens at 50 kg/m^3^ was comparable. At a density of 250 kg/m^3^, the resin-bound panels were 50% more pressure-resistant than their counterparts produced in a wet process. It has to be noted that the CR of resin bound panels was tested at 10% compression, whereas the other specimens were tested at 50% compression.

Wood fiber panels had a compression resistance of 0.04 to 0.2 N/mm^2^ [36], indicating that peat moss panels have adequate pressure resistance for insulation applications. Insulation panels made from miscanthus and a mineral binder showed a CR of 0.3 N/mm^2^ with a density of 250 kg/m^3^ [37], confirming the competitiveness of the peat moss panels.

### 3.6. Water Absorption (WA) and Thickness Swell (TS)

The water absorption ranged from 4 to 60 kg/m^2^, being lowest with light moss mats and highest with heavy wet process panels (Table 5). The effect of both density (*p* = 0.007) and panel type (*p* = 0.000) was statistically significant, although the explanatory power of the panel type is more than twice as high (58%) as that of the density (25%).

The water absorption increased with density. At a density of 100 kg/m^3^ and below, the WA of different specimens was comparable and below 18%. At higher densities, the water absorption was higher with moss mats and wet process panels than with the resin bound samples. The latter showed a maximum water absorption of 20% at 250 kg/m^3^.

A similar coherence was investigated for TS, whose variation was 85%, caused by varying panel type (*p* = 0.000) and density (*p* = 0.000). The explanatory power of the panel type (73%) exceeded that of density (54%). The TS amounted for −26% to 71%. The light moss mats shrunk when wetted. At a density of 50 kg/m^3^, they swelled by 8%, comparable to resin-bound panels at 100 kg/m^3^. Until a density of 100 kg/m^3^, all investigated samples showed a steep increase in TS with rising density (except for the moss mats, which collapsed at low densities when wetted). At higher densities, the wet process panels swelled to disproportionately high levels of 75%, whereas the resin-bound panels showed a lower increase of TS.

Comparing the wet process panels and the resin-bound panels, it is obvious that the resin added to the hydrophobic panel characteristic, because the WA was much lower. This advantage might be relativized in further research, as the resin content will have to be reduced in an industrial environment due to economic reasons. Cotton mats took up 12–13, hemp took up 4.2, extruded polystyrene took up 1.5 to 3 kg/m^2^ water in a soaking test [36]. The resin-bound peat moss and the peat moss mats were comparable to cotton, while the wet process panels showed significantly higher water absorption. The high water uptake of peat moss is caused by the water-storing hyalin cells of peat moss, which in nature are optimized to take up huge amounts of water and might be a source of optimization by adding hydrophobic agents [31].

The thickness swelling of peat moss samples was high compared to a study reporting a TS of 13–20% for rice straw panels, bound with polymeric methylene diphenyl diisocyanate (pMDI), at a density of 250 kg/m^3^. In a study developing insulation materials out of reed mace, the TS after 24 h of water storage was limited with 15% [38]. It has to noted that, in the cited study, the specimens were fully immersed, whereas they were not in the current study. Nonetheless, the benchmark was failed by all moss samples (apart from P1 and M3).

### 3.7. Thermal Conductivity (TC)

The TC of the investigated specimen ranged from 0.038 W/m·K to 0.065 W/m·K (Table 6). This variation is significantly (*p* = 0.001) caused by the specimen density, whereas the effect of the panel type is not significant (*p* = 0.201). A significant part, i.e., 77% (*p* = 0.000), of the variation in TC can be explained by the density. The lowest TC of below 0.04 W/m·K was observed at 50 kg/m^3^, from which TC increased by 0.11 W/m·K per 100 kg/m^3^ increase in density. The light moss mats with a density below 50 kg/m^3^ showed a reverse coherence; TC increased with decreasing density. The total TC of a material is made up by conduction, convection, and radiation in the pores [15]. The latter is relatively high in very light materials (<25 kg/m^3^), which explains the nonlinear relation.

For the investigated temperature range (average measurement temperature 10 to 40 °C), the coefficient of determination between temperature and TC accounted for 99% (*p* = 0.000). The gradient of the linear regression function for the temperature-dependent TC ranged from 0.12–0.18, 0.15–0.18, and 0.13–0.23 mW/m·K per kelvin for the resin-bound panel, wet process panel, and moss mats, respectively. Mineral wool (0.13, 145 kg/m^3^) had a comparable, whereas wood wool (0.31, 348 kg/m^3^) had a significantly higher gradient [16].

The moss samples with a TC between 0.038 and 0.065 W/mK, depending on density, were comparable to natural insulation materials such as wood fibers or flax and synthetic materials such as expanded polystyrene.

### 3.8. Vapor Diffusion Resistance (VDR)

The VDR ranged from 20–22 for the dry test and 4–7 for the wet test (Table 7). The effect of the test type (dry/wet) was significant (*p* = 0.001), whereas that of density was not (*p* = 0.43). Investigating the VDR in the dry test, it increased by 1.9 per 100 kg/m^3^ increase in density. The *R*^2^ between density and VDR accounted for 0.97 in this case. The wet test displayed a different picture with an *R*^2^ of 0.74 between density and VDR, but a rather constant VDR value.

Compared to fiber insulation materials such as flax and hemp mats (VDR 1–2) or wool fiber panels (VDR 2–5) [37], the VDR (dry measured) of peat moss with approximately 20 was high. The fact that the VDR does not rise with density, which was shown for bark insulation panels [39], suggests that the high resin content of 20% functions as a vapor diffusion barrier in the panel. A similar effect was reported for high-density insulation materials [33]. Nevertheless, the peat moss exhibited less vapor tightness than EPS with a VDR of 30–35 [33,40]. Moreover, it was shown in other studies that an exponential relationship exists between density and vapor diffusion resistance. For densities higher than 350 kg/m^3^, a linear relationship exists between the two characteristics [39,41]. Referring to panels with the same density, the VDR was, on average, 75% lower in the wet cup test than in the dry cup test. This is in line with studies focusing on other wood-based panels [39,41].

## 4. Conclusions

The findings show that peat moss is a very promising resource for the production of insulation materials. It is suitable for both flexible mats and panels. The production process for flexible mats will have to be optimized to enable industrial production. The insulation materials presented in this study exhibited some weaknesses. The lower-density panels showed low mechanical stability. This limits their application in non-load-bearing situations. The resin-bound panels incorporated a high amount of synthetic resin. Further research will have to address resin reduction and its replacement by more ecological materials. A reduction in resin content might reduce the high vapor diffusion resistance. Other questions arising from the current paper are the moisture-dependent thermal conductivity of moss and its durability against fungal and pest infestation, as well as fire resistance.

The peat moss specimens showed a low thermal conductivity comparable to other light insulation materials. Their moisture sorption and water absorption characteristics limit their application to interior or water-shaded situations. However, they also offer potential; due to the high hygroscopicity of moss, the insulation panels/mats are an ideal choice where moisture-active materials are required (e.g., compensation of temporary condensation with interior insulation layers [42]).

## Figures and Tables

**Figure 1 materials-14-06601-f001:**
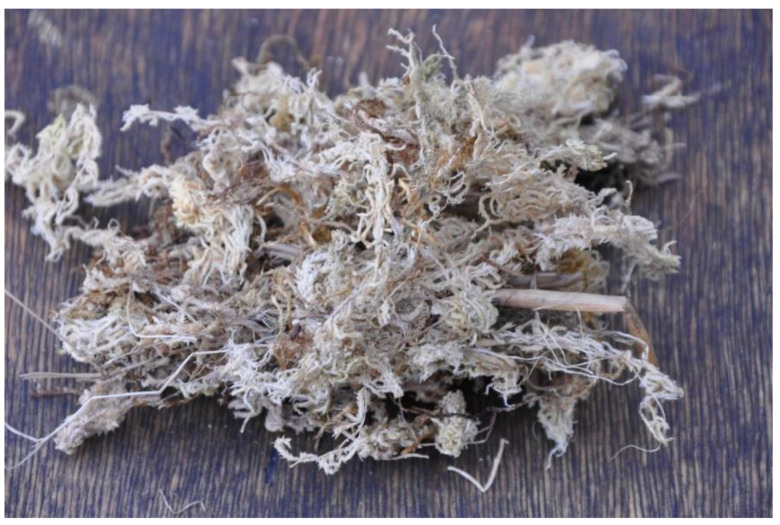
Dry peat moss (sphagnum).

**Figure 2 materials-14-06601-f002:**
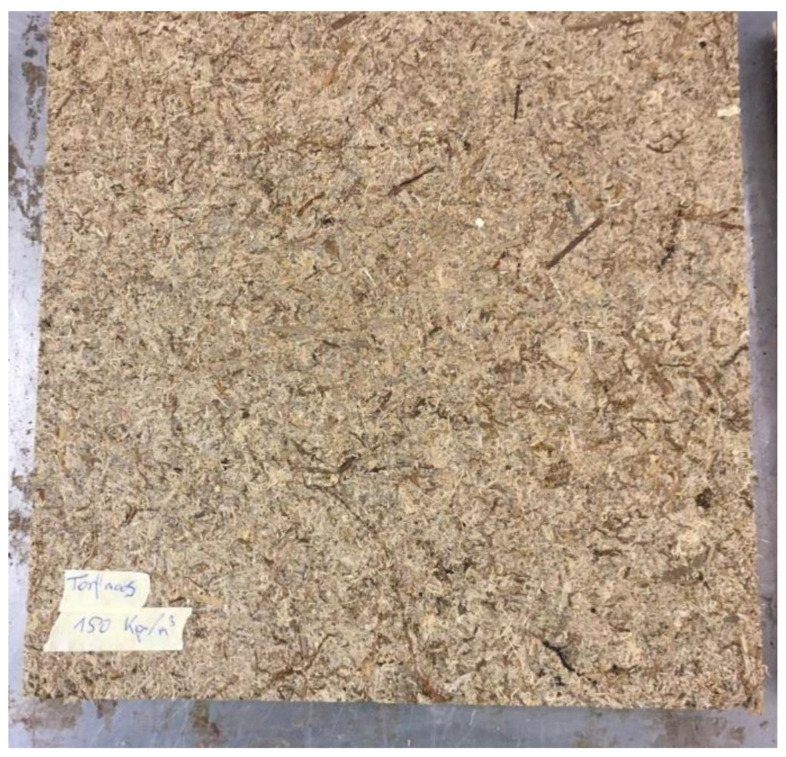
Peat moss panel (resin-bound) with a density of 150 kg/m^3^.

**Figure 3 materials-14-06601-f003:**
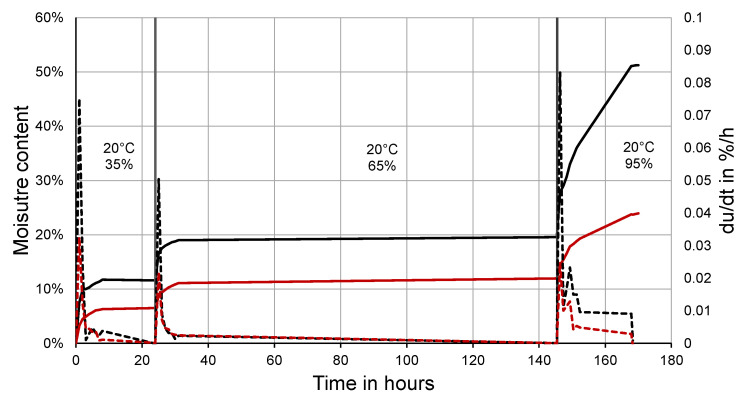
Hygroscopic response of peat moss and saw dust to different climates with sorption rates of the moss and wood saw dust (du/dt = relative moisture uptake per time unit).

**Table 1 materials-14-06601-t001:** DOE for the production of peat moss specimens.

Panel Type	Resin Content	Target Density	Replicates
	(%)	(kg/m^3^)	
P1	20	100	2
P2	20	150	2
P3	20	250	2
WP1	-	50	2
WP2	-	150	2
WP3	-	250	2
M1	-	15	2
M2	-	20	2
M3	-	50	2

P = resin-bound panel, WP = wet process panel, M = moss mat.

**Table 2 materials-14-06601-t002:** Overview of MOR and MOE testing (SD in brackets).

Panel Type	Target Density	MOR	MOE	Number
	(kg/m^3^)	(N/mm^2^)	(N/mm^2^)	
P1	100	0.09 (0.03)	4.95 (1.79)	5
P2	150	0.20 (0.06)	15.96 (7.95)	5
P3	250	1.45 (0.33)	114.59 (27.06)	5
WP2	150	0.08 (0.02)	2.70 (0.34)	11
WP3	250	0.10 (0.01)	2.54 (0.48)	4

**Table 3 materials-14-06601-t003:** Internal bond of moss samples (SD in brackets).

Panel Type	Target Density	IB	Number
	(kg/m^3^)	(N/mm^2^)	
P1	100	0.003 (0.001)	1
P2	150	0.003 (0.001)	5
P3	250	0.019 (0.010)	5
WP2	100	0.001 (-)	2
WP3	150	0.002 (0.001)	5
M1	15	0.019 (0.006)	5
M2	20	0.027 (0.008)	5
M3	50	0.026 (0.002)	5

**Table 4 materials-14-06601-t004:** Compression resistance of moss samples (SD in brackets).

Panel Type	Target Density	Compression	CR	Number
	(kg/m^3^)	(%)	(N/mm^2^)	
P1	100	10	0.05 (0.01)	2
P2	150	10	0.14 (0.07)	2
P3	250	10	0.50 (0.06)	2
WP1	50	50	0.05 (0.01)	2
WP2	150	50	0.15 (0.04)	2
WP3	250	50	0.33 (0.02)	2
M1	15	50	0.01 (0.00)	4
M2	20	50	0.02 (0.00)	4
M3	50	50	0.04 (0.01)	4

**Table 5 materials-14-06601-t005:** Water absorption and thickness swell of peat moos samples (SD in brackets).

Panel Type	Target Density	WA	TS	Number
	(kg/m^3^)	(kg/m^2^)	(%)	
P1	100	11.96 (0.84)	8.3 (2.8)	3
P2	150	13.84 (1.46)	23.2 (0.2)	3
P3	250	19.77 (0.08)	44.5 (1.3)	3
WP1	50	12.88 (2.15)	15.0 (0.00)	2
WP2	150	34.99 (3.83)	71.0 (8.5)	2
WP3	250	60.75 (1.35)	66.0 (4.2)	2
M1	15	4.20 (0.69)	−22.0 (4.7)	5
M2	20	4.07 (0.66)	−25.8 (4.8)	5
M3	50	16.99 (4.26)	8.4 (7.1)	5

**Table 6 materials-14-06601-t006:** Thermal conductivity (SD in brackets).

Panel Type	Target Density	TC	Number
	(kg/m^3^)	(W/mK)	
P1	100	0.043 (-)	1
P1	150	0.048 (-)	1
P2	250	0.065 (-)	1
WP1	50	0.039 (0.001)	2
WP2	100	0.044 (-)	1
WP3	150	0.046 (-)	1
M1	15	0.043 (0.000)	2
M2	20	0.041 (0.001)	2
M3	50	0.038 (0.001)	2

**Table 7 materials-14-06601-t007:** Vapor diffusion resistance of resin-bound peat moss panels.

Panel Type	Target Density	Test Type	VDR	Number
	(kg/m^3^)	(kg/m^2^)	(-)	
P1	100	dry	21.47	1
P1	100	wet	4.27	1
P2	150	dry	21.78	1
P2	150	wet	4.74	1
P3	250	dry	20.32	1
P3	250	wet	6.99	1

## Data Availability

Data sharing not applicable.

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
