# Peer review of "Production and Physical–Mechanical Characterization of Peat Moss (Sphagnum) Insulation Panels"

_materials, 2021, doi:10.3390/ma14216601_

Round 1

Reviewer 1 Report

The article is well organized and written. The aim of this article is to conduct basic research on modified and unmodified peat moss, which allows them to be used as filling and insulation.. Comments on the text below:

  1. explain the notations in Table 1
  2. provide the methodology of Internal bond (IB) determination
  3. check the literature for correctness
  4. What resins can be used. Can you use recycled resins?
  5. What materials can be used instead of resin.
  6. What would be the durability (corrosion) of such materials?
  7. Under the conditions, the authors see the use of these materials?

Author Response

Reviewer 1

English language and style

( ) Extensive editing of English language and style required

( ) Moderate English changes required

(x) English language and style are fine/minor spell check required

( ) I don't feel qualified to judge about the English language and style

Yes        Can be improved            Must be improved         Not applicable

Does the introduction provide sufficient background and include all relevant references?

(x)         ( )          ( )          ( )

Is the research design appropriate?

(x)         ( )          ( )          ( )

Are the methods adequately described?

( )          (x)         ( )          ( )

Are the results clearly presented?

(x)         ( )          ( )          ( )

Are the conclusions supported by the results?

(x)         ( )          ( )          ( )

Comments and Suggestions for Authors

The article is well organized and written. The aim of this article is to conduct basic research on modified and unmodified peat moss, which allows them to be used as filling and insulation. Comments on the text below:

explain the notations in Table 1

Notations were explained in a table footer.

provide the methodology of Internal bond (IB) determination

A short description of IB testing was implemented in the text.

check the literature for correctness

Literature resources haven been checked.

What resins can be used. Can you use recycled resins?

All sort of resins can be used, resulting in differing panel properties. A follow-up study will be presented soon focusing on exactly this issue. Regarding recycled resins, basically it is no problem using them. Which one do you refer to?

What materials can be used instead of resin.

It is a basic question, which strategy should be applied to bind moss particles. We have tried to suggest three of them (binder less, tissue, resin). There is also historic evidence that spinning technology was used to form moss objects. As the resulting panels/mats differ greatly in their technical properties, the type of application will be the most important factor regarding this question.

What would be the durability (corrosion) of such materials?

Further experimental work will have to be conducted on the natural durability of moss against fungi and other biological attack. For tendency, the durability is better than reference materials (e.g. wood fibers) because the moos is quite acid and lacks a high amount of sugar components. The need of further research in this direction was indicated in the “Conclusions”.

Under the conditions, the authors see the use of these materials?

We see great potential for the material, due to its sustainable resource basis and a lack of existing high-value applications. Moreover, there is quite a bit of historic evidence for successful use of peat moss in buildings. The current work shall lay the basis for modern peat moss applications.

Reviewer 2 Report

The manuscript deals with the production and characterization of peat moss insulation panels. The manuscript has wide enough and informative introduction. The materials and methods are clearly presented - however some detail information could be presented directly in this chapter, and not just refered to previous research or standards (i.e. vapor diffusion resistance). Results are clearly presented - no redundancy has been found. The discussion is broad and deep enough. Conclusions are well supported by the findings.

Remarks / comments:

L97 Please correct the unit for thermal conductivity (TC) into mW/m2K

L153 Please correct unit for temperature into °C

L210 Please explain the value du/dt on secondary axis

L211 It is suggested to rewrite the caption of the figure 2. The sorption curve is usually meant the graph of equilibrium moisture content vs. relative humidity. In your case it looks more like a hygroscopic response or behaviour of the tested material.

L294 Please describe the abbreviation PMDI

L302 The sentence starts with number: "77 % (p=0.000) of the variation..." It needs rewording. Check the grammar!

L315 "The moss samples having a TC between 0.038 and 0.065,..." Please add the units for TC.

L318 There is not correct unit in Table 6 for TC! Is it really %?

L337 Please check the symbol VR and unit (%) in table 7! Isn't there proper VDR?

Author Response

Reviewer 2

Open Review

(x) I would not like to sign my review report

( ) I would like to sign my review report

English language and style

( ) Extensive editing of English language and style required

( ) Moderate English changes required

(x) English language and style are fine/minor spell check required

( ) I don't feel qualified to judge about the English language and style

Yes        Can be improved            Must be improved         Not applicable

Does the introduction provide sufficient background and include all relevant references?

(x)         ( )          ( )          ( )

Is the research design appropriate?

(x)         ( )          ( )          ( )

Are the methods adequately described?

(x)         ( )          ( )          ( )

Are the results clearly presented?

(x)         ( )          ( )          ( )

Are the conclusions supported by the results?

(x)         ( )          ( )          ( )

Comments and Suggestions for Authors

The manuscript deals with the production and characterization of peat moss insulation panels. The manuscript has wide enough and informative introduction. The materials and methods are clearly presented - however some detail information could be presented directly in this chapter, and not just refered to previous research or standards (i.e. vapor diffusion resistance). Results are clearly presented - no redundancy has been found. The discussion is broad and deep enough. Conclusions are well supported by the findings.

The determination of the vapor diffusion resistance was described in short in the methods part of the paper.

Remarks / comments:

L97 Please correct the unit for thermal conductivity (TC) into mW/m2K

Thank you, but the unit of the temperature dependent rise of TC is mW/mK*1/K=mW/mK². Indeed, this is confusing, so we improved the sentence.

L153 Please correct unit for temperature into °C

Thank you – mistake was corrected.

L210 Please explain the value du/dt on secondary axis

The figure footer was supplemented by adding “(du/dt=relative moisture uptake per time unit)”

L211 It is suggested to rewrite the caption of the figure 2. The sorption curve is usually meant the graph of equilibrium moisture content vs. relative humidity. In your case it looks more like a hygroscopic response or behaviour of the tested material.

Thank you this is an important aspect to avoid misunderstanding. Your suggestion was implemented in the caption.

L294 Please describe the abbreviation PMDI

“Polymeric methylene diphenyl diisocyanate” was named.

L302 The sentence starts with number: "77 % (p=0.000) of the variation..." It needs rewording. Check the grammar!

The sentence was improved.

L315 "The moss samples having a TC between 0.038 and 0.065,..." Please add the units for TC.

Units were inserted.

L318 There is not correct unit in Table 6 for TC! Is it really %?

It is of course “W/mK” – thank you for your attentive reading.

L337 Please check the symbol VR and unit (%) in table 7! Isn't there proper VDR?

The symbol was changed to VDR and the unit to (-). The vapor diffusion resistance is a comparison value and carries no unit.

Reviewer 3 Report

• Addition of key finding result in the abstract need to be added to captures the reader’s interest and imagination. • Addition of a few illustrations especially where the applications are concerned to (in very specific way) capture the readers interest and imaginations, in the introduction section. • Future trends and prospects need to be elaborated further, stipulating the research directions. • Under experimental section, author need to be describe all the used chemicals with their manufacturer names, purity and place of manufacturing materials under material section. • Author need to present his result in figure format not in the tabular one.

Author Response

Reviewer 3

Open Review

(x) I would not like to sign my review report

( ) I would like to sign my review report

English language and style

( ) Extensive editing of English language and style required

( ) Moderate English changes required

( ) English language and style are fine/minor spell check required

(x) I don't feel qualified to judge about the English language and style

Yes        Can be improved            Must be improved         Not applicable

Does the introduction provide sufficient background and include all relevant references?

( )          (x)         ( )          ( )

Is the research design appropriate?

( )          (x)         ( )          ( )

Are the methods adequately described?

( )          (x)         ( )          ( )

Are the results clearly presented?

( )          ( )          (x)         ( )

Are the conclusions supported by the results?

(x)         ( )          ( )          ( )

Comments and Suggestions for Authors

Addition of key finding result in the abstract need to be added to captures the reader’s interest and imagination.

The most important findings were incorporated in the abstract.

Addition of a few illustrations especially where the applications are concerned to (in very specific way) capture the readers interest and imaginations, in the introduction section.

An illustration of a historic sealing cord was incorporated to the introductory part to capture the readers interest and imagination.

Future trends and prospects need to be elaborated further, stipulating the research directions.

Future research directions were discussed in the “Conclusions”.  

Under experimental section, author need to be describe all the used chemicals with their manufacturer names, purity and place of manufacturing materials under material section.

The manufacturer name and technical details of the resin used were supplied. No further chemicals were used.

Author need to present his result in figure format not in the tabular one.

We have considered to present results as figures, e.g., box plots, but came to the conclusion that short tables might be more precise. We suggest keeping the tables for this reason.